# *AutoAWG*: Adverse Weather Generation with Adaptive Multi-Controls for Automotive Videos

## Abstract

Perception robustness under adverse weather remains a critical challenge for autonomous driving, with the core bottleneck being the scarcity of real-world video data in adverse weather. Existing weather generation approaches struggle to balance visual quality and annotation reusability. We present *AutoAWG*, a controllable *A*dverse *W*eather video *G*eneration framework for *Auto*nomous driving. Our method employs a semantics-guided adaptive fusion of multiple controls to balance strong weather stylization with high-fidelity preservation of safety-critical targets; leverages a vanishing point-anchored temporal synthesis strategy to construct training sequences from static images, thereby reducing reliance on synthetic data; and adopts masked training to enhance long-horizon generation stability. On the nuScenes validation set, *AutoAWG* significantly outperforms prior state-of-the-art methods: without first-frame conditioning, FID and FVD are relatively reduced by 50.0% and 16.1%; with first-frame conditioning, they are further reduced by 8.7% and 7.2%, respectively. Extensive qualitative and quantitative results demonstrate advantages in style fidelity, temporal consistency, and semantic–structural integrity, underscoring the practical value of *AutoAWG* for improving downstream perception in autonomous driving.

## 1 Introduction

Perception robustness under adverse weather (e.g., nighttime, rain, snow, fog) is a key challenge for autonomous driving, and its core bottleneck lies in the extreme scarcity of real-world data (Zhang et al., 2023). Among existing solutions, weather removal methods (Ni et al., 2021; Valanarasu et al., 2022; Yang et al., 2024b) are difficult to deploy due to real-time constraints, while weather generation methods (Li et al., 2022; Lan et al., 2024; Zhao et al., 2024b) often fail to preserve the original scene structure, making annotations non-reusable and thus costly. Video weather style transfer offers a practical alternative: it can synthesize diverse weather conditions while maximally preserving original annotations, providing efficient and low-cost data augmentation for perception models. However, this task imposes dual stringent requirements: it must produce highly realistic weather appearances and simultaneously preserve the geometry and semantics of safety-critical objects.

Based on this, we argue that an effective model for autonomous-driving video weather style transfer should possess two core capabilities: (1) **style fidelity and temporal consistency** — faithfully reproducing the visual characteristics of the target weather and maintaining consistent style across consecutive frames; and (2) **semantic-structural consistency** — precisely preserving the semantics and geometry of safety-critical objects (vehicles, pedestrians, traffic signs) under domain shifts, over time, and across multi-camera views, thereby ensuring that the translated videos remain usable for downstream perception tasks.

To enforce semantic–structural consistency, prior work commonly introduces high-level controls such as 3D bounding boxes, BEV maps, or trajectory maps (Gao et al., 2023; 2024; Wang et al., 2024b; Chen et al., 2024; Zhao et al., 2025; Xie et al., 2025). While these representations help ensure multi-view and temporal geometric consistency, they lack fine-grained guidance for textures and local structures, leading to limited visual detail and under-expressive or less realistic weather effects (e.g., overly bright night scenes). On the other hand, to secure style fidelity and consistency,

many approaches rely heavily on paired data for supervision (Zhou et al., 2024b; Lin et al., 2025; Song et al., 2025). However, acquiring such paired data of the same scene under multiple weather conditions is practically infeasible in real world. As a workaround, synthetic data are often used for training, but they introduce non-trivial domain gaps, and the synthesis pipeline itself is costly and prone to artifacts and geometric/texture biases.

To address these limitations from a structure–style decoupling perspective, we propose *AutoAWG*. Our key insight is to introduce a set of complementary control conditions and to conceptualize their fusion as a "coloring-book" process, which tackles the challenges of control granularity and data dependence. Concretely, we formulate video weather transfer as generation guided by structural priors: Lineart outlines objects' boundaries and shapes; Depth and Sketch jointly define the global scene structure and layering; and a semantic segmentation mask partitions the canvas into distinct coloring regions. Building on this foundation, our semantics-guided adaptive fusion and importance-weighted loss effectively bold the contours for safety-critical regions (e.g., vehicles, pedestrians). Under such constraints, the diffusion model is simplified to filling appropriate colors and textures for each region according to a target-weather palette. This explicit disentanglement of style (coloring) from content (structural sketch) ensures strong stylization while preserving object integrity.

To address data scarcity, we further propose a vanishing point-anchored temporal synthesis: by keeping the normalized location of the vanishing point fixed, we generate an equal-ratio cropping sequence, resize all crops to a common resolution, and concatenate them along the temporal dimension to form a pseudo-video that simulates stable forward motion from still images. This substantially mitigates the scarcity of adverse-weather videos and the domain gaps introduced by synthetic pipelines. Finally, to support long-horizon video generation, we adopt a masked segmented training strategy. By either randomly masking all frames or keeping only the first frame while masking the rest, the model is compelled to learn long-range temporal dependencies, ensuring indefinite continuation and temporal consistency in the generated videos.

**Core Advantages of *AutoAWG* : Three Highs**

- **High-Quality Generation.** With semantics-guided adaptive fusion of multiple controls and importance weighting, the model dynamically allocates controls to different regions, striking a balance between strong weather stylization and high-fidelity objects preservation.
- **High Consistency and Reusability.** By directly leveraging pixel-level controls extracted from the input videos, *AutoAWG* keeps object geometry and semantics tightly aligned with the original scene. The translated videos can therefore reuse existing annotations (e.g., 2D/3D labels, LiDAR) without re-annotation, enabling plug-and-play integration into downstream tasks.
- **High Flexibility and Scalability.** Thanks to robust controllability and training strategies, the framework naturally supports multi-camera systems and arbitrary-length sequences, meeting practical requirements while reducing cross-frame fluctuation.

## 2 RELATED WORK

### 2.1 ADVERSE WEATHER IN AUTONOMOUS DRIVING

**Adverse Weather Removal.** Many methods aim to enhance perception in adverse weather by restoring clear visual cues. Initial studies focused on single-condition image restoration (e.g., de-raining, de-snowing, de-hazing) (Ni et al., 2021; Valanarasu et al., 2022; Özdenizci & Legenstein, 2023), while more recent works extend to multi-condition video restoration (Yang et al., 2023; 2024b). These models are often trained on synthetic datasets such as Outdoor-Rain (Li et al., 2019), RainDrop (Qian et al., 2018), and Snow100K (Liu et al., 2018), though recent efforts are shifting toward real-world data (Zhu et al., 2023). A comprehensive review is provided in (Xiao et al., 2024).

**Adverse Weather Generation.** Another direction involves generating adverse weather data to augment training datasets and improve robustness under challenging conditions. Early methods used GANs to synthesize weather effects (Kwak et al., 2021; Lan et al., 2024; Li et al., 2022), but their instability has led to interest in rendering-based approaches (Wang et al., 2023; Zhao et al., 2024b). However, most of these works are image-based, making them difficult to extend to videos and limiting their utility in sequential tasks like autonomous driving.

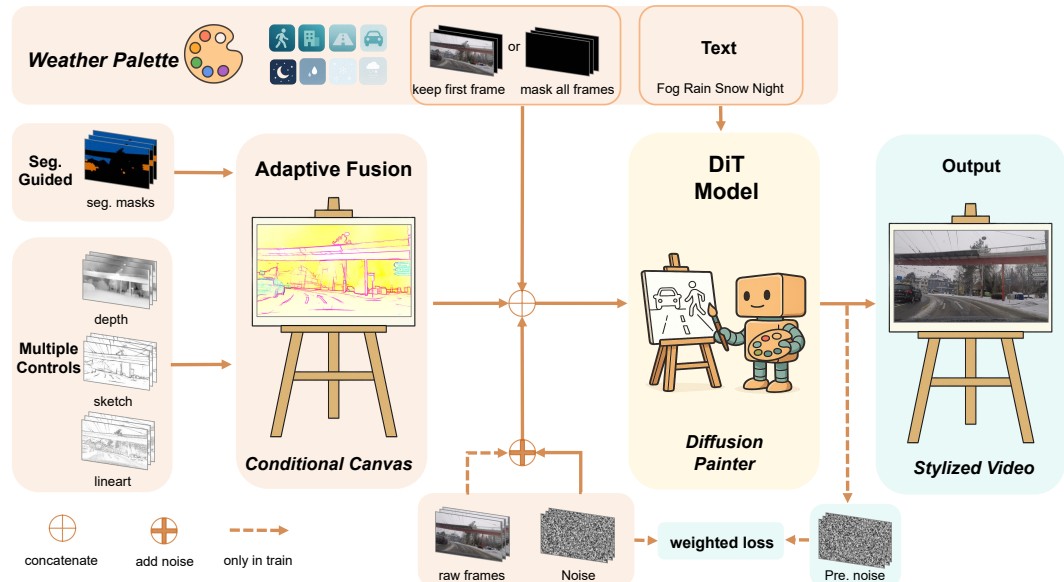

Figure 1: Overview of the proposed *AutoAWG* for adverse weather generation.

While recent methods such as Panacea (Wen et al., 2024) and UniMLVG (Chen et al., 2024) can generate driving videos under various weather conditions, they have notable limitations. The former relies on an image-to-image translation model to convert the initial frame, whereas the latter requires pre-training on large-scale, web-crawled data with complex labeling. Moreover, none of these methods explicitly considers the preservation of scene details, making the generated videos unsuitable for direct reuse of existing annotations.

## 2.2 VIDEO GENERATION IN AUTONOMOUS DRIVING

Recent studies on generation for autonomous driving can be broadly categorized into two paradigms: reconstruction-based approaches and controllable generation approaches.

Reconstruction-based methods aim to regenerate the 3D driving scene using multi-view images, often aided by LiDAR data. These include techniques based on NeRF (Mildenhall et al., 2021) and 3D Gaussian Splatting (Kerbl et al., 2023), such as (Yan et al., 2024; Zhou et al., 2024a), which reconstruct detailed dynamic environments from onboard sensors. Controllable generation methods, on the other hand, utilize diffusion models guided by structured conditions such as camera trajectories, BEV maps, or textual prompts (Gao et al., 2023; Wang et al., 2024a;b; Wen et al., 2024; Zhou et al., 2024b; Wu et al., 2025; Chen et al., 2024; Zhao et al., 2025; Ni et al., 2025). Some methods integrate reconstruction and generation (Ni et al., 2024; Zhao et al., 2024a).

Our work falls into the controllable generation category, with a focus on transforming existing autonomous driving scenes into adverse weather conditions. Unlike methods that synthesize entirely new scenes, our approach preserves the original layout and annotations, enabling direct usage in downstream perception tasks without the need for re-labeling.

## 3 METHOD

### 3.1 ARCHITECTURAL OVERVIEW

We formulate adverse weather generation as a video style transfer problem within a controllable diffusion framework. As illustrated in Figure 1, multiple complementary control conditions are adaptively fused according to semantic masks to construct a conditional canvas. The DiT model then acts as a "Painter," filling this canvas with realistic weather effects while preserving object fidelity.

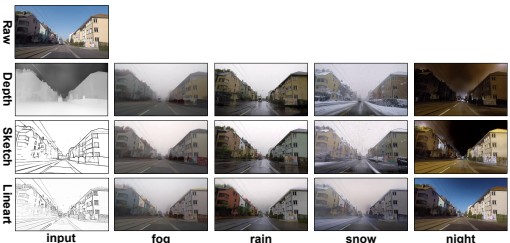

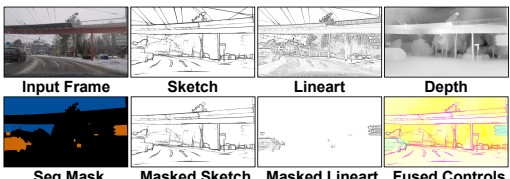

Figure 2: Comparison of different control maps and generated results.

Figure 3: Adaptive fusion of controls guided by segmentation masks.

Given an input video, multiple structural controls are first extracted and safety-critical objects are segmented. During training, frames are encoded into the latent space via a 3D VAE, perturbed with Gaussian noise, and progressively denoised through the DiT. At inference, generation starts from random noise, guided by a target weather embedding and the fused conditions, to produce a temporally consistent transformed sequence. Detailed descriptions of each component are provided in the following sections.

### 3.2 ADAPTIVE FUSION OF MULTIPLE CONTROLS

We propose a region-aware adaptive fusion strategy to integrate multiple control maps of varying strengths.

**Control Strength Spectrum.** We observe that different control conditions vary in how strongly they relate to the source content. Some control conditions capture a fine details, making it easy to generate content that closely resembles the original imagery but difficult to adapt to other styles. In contrast, others extract only coarse structural outlines, making it harder to reconstruct the original content details but facilitating strong style transformation.

As shown in Figure 2, the depth map (Yin et al., 2021) captures only coarse structural layouts, which facilitates strong transformations into adverse weather conditions. In contrast, the lineart map retains fine-grained details, even down to textures such as tree bark, thereby constraining large stylistic changes. The sketch map (Su et al., 2023) lies between these two extremes, achieving a balance between content preservation and style flexibility. This trend is further corroborated by the results in Table 4.

**Region-Aware Control Fusion.** Building on these findings, we selectively combine control maps using semantic masks. Critical objects (e.g., vehicles, pedestrians, traffic signs) must remain consistent, so only their regions are retained in the lineart map. Sky regions are removed from sketch maps to allow flexible weather variations, while depth maps are preserved entirely. The fused control is formulated as:

$$\mathbf{C}_a = \text{concat}\{\mathbf{C}_d, \mathbf{C}_l \odot \mathbf{M}_{obj}, \mathbf{C}_s \odot (\mathbf{1} - \mathbf{M}_{sky})\}, \tag{1}$$

where $\mathbf{C}_d, \mathbf{C}_l, \mathbf{C}_s$ denote depth, lineart, and sketch maps, and $\mathbf{M}_{obj}, \mathbf{M}_{sky}$ are object and sky masks. Figure 3 illustrates this segmentation-aware multi-control fusion. Fused controls are then combined into a video and encoded via the 3D VAE.

### 3.3 IMPORTANCE-WEIGHTED LOSS

We adopt the Flow Matching framework (Lipman et al., 2023). For a latent video sample $\mathbf{X}_1$ and Gaussian noise $\mathbf{X}_0 \sim \mathcal{N}(\mathbf{0}, \mathbf{1})$, the interpolated state is $\mathbf{X}_t = t\mathbf{X}_1 + (1-t)\mathbf{X}_0$. The model predicts the velocity $\mathbf{V}_t$ to approximate the ground truth $\mathbf{U}_t = d\mathbf{X}_t/dt$. Standard training minimizes:

$$\mathcal{L} = \mathbb{E}\|\mathbf{U}_t - \mathbf{V}_t\|^2. \tag{2}$$

However, uniform weighting overlooks varying regional importance. To emphasize critical regions, we introduce an importance-weighted loss:

$$\mathcal{L} = \mathbb{E}\left[\|(\mathbf{U}_t - \mathbf{V}_t)\|^2 + \alpha \cdot \|\mathbf{M}_{obj} \odot (\mathbf{U}_t - \mathbf{V}_t)\|^2\right], \tag{3}$$

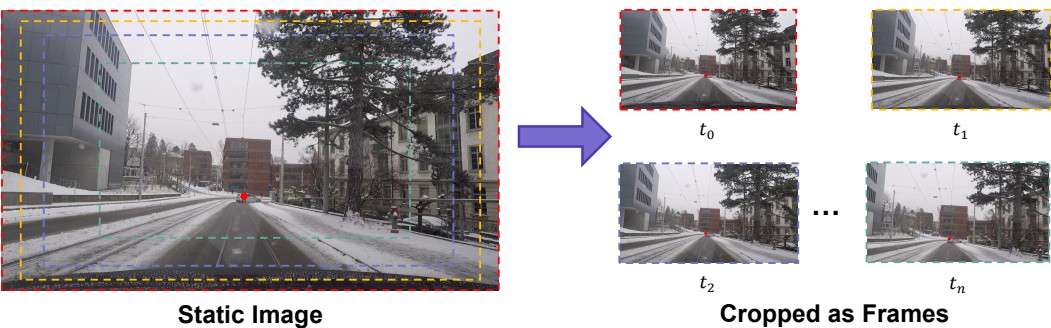

**Stitched Multi-Camera**                    **Multi-Camera's Control Maps**

Figure 4: Multi-camera scenario: stitched frames and corresponding controls.

**Static Image**                    **Cropped as Frames**

Figure 5: VP-Anchored Temporal Synthesis: synthesize video from a single image.

where $\mathbf{M}_{obj}$ denotes object masks and $\alpha$ controls relative importance. This prioritizes accurate reconstruction of critical objects such as vehicles and pedestrians.

### 3.4 MULTI-VIEW AND UNLIMITED-LENGTH GENERATION

Our framework naturally extends to multi-camera setups in autonomous driving. Since control maps operate at the pixel level, spatially aligned and temporally synchronized inputs ensure consistent generation across views. We stitch frames from all cameras into a single composite grid and apply the same operation to their corresponding control maps (Figure 4), ensuring uniform appearance of shared objects across cameras. Compared to prior methods that rely on explicit cross-view attention (Gao et al., 2023; Wen et al., 2024; Chen et al., 2024), our approach is simpler yet effective. The spatial concatenated multi-view frames are treated as a joint input, which is then partitioned into tokens within the DiT architecture. Through self-attention, DiT models the relationships among these tokens, enabling our approach to implicitly capture the correspondences and consistency across different camera views.

For unlimited-length sequences, we adopt a segment-wise strategy using specific inpainting masks. During training, we randomly mask either all frames (for generating the first segment) or all but the first frame (for continuation generation). At inference, each new segment is conditioned on its control maps and the last frame of the previous segment (none for the first segment), ensuring temporal continuity over arbitrarily long sequences.

### 3.5 ADVERSE WEATHER TRAINING DATA

To compensate for the lack of open-source diverse adverse-weather driving videos, we construct a mixed training set from image and video datasets. Specifically, we propose a crop-to-video strategy to convert ACDC images to videos, and use them together with nuScenes samples to train our model.

**ACDC Dataset.** ACDC (Sakaridis et al., 2021) contains 4,006 images across fog, rain, snow, and nighttime. As it lacks temporal sequences, we propose the VP(Vanishing Point)-Anchored Temporal Synthesis strategy to synthesize videos from static images. Specifically, we first estimate the vanishing point (Pautrat et al., 2023) in the image and then continuously crop the image using a fixed aspect ratio, ensuring that the vanishing point's relative position remains unchanged in each cropped image. The original image resolution is 1920×1080, and the final cropped image is fixed

Table 1: Quantitative comparison with automotive video generation methods on nuScenes dataset.

| Methods | w/ 1st frame condition | FID↓ | FVD↓ |
|---|---|---|---|
| MagicDrive (Gao et al., 2023) | ✗ | - | 217.9 |
| MagicDrive-V2 (Gao et al., 2025) | ✗ | - | 94.8 |
| DriveDreamer (Wang et al., 2024a) | ✗ | 26.8 | 353.2 |
| DriveDreamer-2 (Zhao et al., 2025) | ✗ | 25.0 | 105.1 |
| DiVE (Jiang et al., 2024) | ✗ | - | 94.6 |
| Ours | ✗ | **12.5** | **79.4** |
| GenAD (Yang et al., 2024a) | ✓ | 15.4 | 244 |
| Drive-WM (Wang et al., 2024b) | ✓ | 15.8 | 122.7 |
| Panacea (Wen et al., 2024) | ✓ | 16.9 | 139.0 |
| Vista (Gao et al., 2024) | ✓ | 6.9 | 89.4 |
| DriveDreamer-2 (Zhao et al., 2025) | ✓ | 11.2 | 55.7 |
| GEM (Hassan et al., 2025) | ✓ | 10.5 | 158.5 |
| Glad (Xie et al., 2025) | ✓ | 11.2 | 188.0 |
| DriveScape (Wu et al., 2025) | ✓ | 8.3 | 76.4 |
| MaskGWM (Ni et al., 2025) | ✓ | 8.9 | 65.4 |
| Ours | ✓ | **6.3** | **51.7** |

at 960×544, with the resolutions of the intermediate images decreasing uniformly. Finally, all the cropped images are resized to the same resolution and concatenated along the temporal dimension to produce a video. After that, each image in ACDC is thus transformed into a 45-frame pseudo-video simulating driving motion. Figure 5 demonstrated this process.

**nuScenes Dataset.** nuScenes (Caesar et al., 2020) provides 1,000 scenes of 20-second multi-camera videos at 12 FPS, annotated with 3D bounding boxes. Although only nighttime and rainy conditions are included, its large-scale, multi-view setting makes it an ideal complement to ACDC-derived videos. Together, they form a comprehensive dataset for training adverse weather generation.

## 4 EXPERIMENTS

### 4.1 EVALUATION METRICS

We evaluate our method from two aspects: the success of weather generation and the preservation of original scene content. For the weather alignment, we use CLIP (Radford et al., 2021) to classify each generated frame according to weather (sunny, rainy, foggy, snowy) or time (daytime or nighttime). The average classification accuracy is reported as *Weather* Score. To asses the content preservation, we employ the widely used FID (Heusel et al., 2017) and FVD (Unterthiner et al., 2018). Additionally, we use mAP from object detection like several prior works (Gao et al., 2023). Specifically, we apply the YOLO11X [1] to detect traffic elements in the transformed frames, and compute mAP using the COCO evaluation protocol (Lin et al., 2014).

### 4.2 IMPLEMENTATION DETAILS

We train our model using 8 NVIDIA H20 GPUs. All experiments were conducted in a 45-frame configuration, with the single-camera resolution fixed at 960×544. Firstly, we use the nuScenes dataset for initial training to capture the patterns of autonomous driving scenes. Subsequently, we finetune the model on a combined dataset consisting of nuScenes and the ACDC synthesized video, so that it can learn adverse weather effects. For multi-camera scenarios, we further finetune the model on nuScenes 6-camera videos.

For the ACDC dataset, we use the annotations for "human", "vehicle", "traffic light", "traffic sign" to construct the critical object mask $\mathbf{M}_{obj}$, and the "sky" to build the sky mask $\mathbf{M}_{sky}$. For frames in the nuScenes dataset without segmentation masks, we use DeepLabv3 (Chen et al., 2017) pretrained

---

[1] https://docs.ultralytics.com/models/yolo11/

Table 2: Impact of generated data on BEVFusion 3D object detection on nuScenes dataset.

| Methods | mAP↑ | NDS↑ |
|---|---|---|
| w/o gen. data | 35.53 | 41.20 |
| w/ gen. data | **37.52**+1.99 | **42.56**+1.36 |

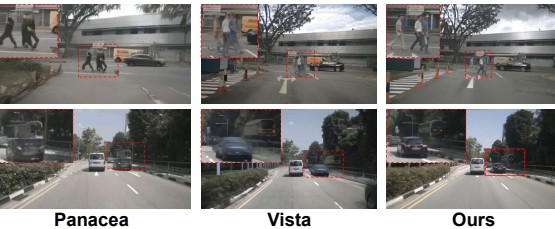

**Panacea**      **Vista**      **Ours**

Figure 6: Visual comparison with Panacea (Wen et al., 2024) and Vista (Gao et al., 2024). Our approach generates more realistic and detailed vehicles, pedestrians, and obstacles compared to Panacea and Vista.

Figure 7: FID and FVD for long video generations. Our method achieves lower and more stable scores than GEM (Hassan et al., 2025) and Vista (Gao et al., 2024).

on the Cityscapes dataset (Cordts et al., 2016) to obtain these masks. We use the controlnet-aux toolboxto extract all the control conditions.

We adopt CogVideoX1.5-5B (Yang et al., 2025) as our backbone, and use its 3D VAE to encode the fused control maps, rather than training an encoder from scratch.

### 4.3 QUANTITATIVE EVALUATION

**Generation Quality.** To assess the visual quality of generated automotive videos, we compare our method against several state-of-the-art approaches in the field of autonomous driving video generation. Following the evaluation protocols in prior works (Wang et al., 2024a;b; Zhao et al., 2025), we conduct quantitative analysis on the nuScenes validation set. To ensure a fair comparison, we configure our model to transform each input video into the same weather condition as its original, rather than into an adverse condition. This setup allows us to focus purely on evaluating the generative quality of the scene. As shown in Table 1, our method achieves an FID of 12.5 and FVD of 79.4 without the first-frame input, and further improves to an FID of 6.3 and FVD of 51.7 when conditioned on the first frame, both substantially surpassing previous state-of-the-art methods. These results highlight the strong ability of our framework to generate high-quality automotive videos.

**Downstream Utility Evaluation.** To evaluate the practical value of our generated videos, we examine their effectiveness in enhancing downstream perception tasks. Specifically, we augment the nuScenes training set with our generated frames (limited to sunny, rainy, and nighttime scenes to match the validation distribution), and use this combined dataset to train a camera-only BEVFusion model (Liu et al., 2023) for 3D object detection. As shown in Table 2, incorporating our synthetic data leads to noticeable improvements: the model's mAP increases by 1.99 points, and the NDS improves by 1.36 points. These results indicate that our generated videos not only exhibit high visual quality but also offer tangible benefits for real-world perception algorithms in autonomous driving.

**Edit Fidelity.** We evaluate edit fidelity following Gao et al. (2025) by transferring validation cases to normal, rainy, and nighttime conditions while retaining the same ground-truth annotations as raw frames. The pretrained BEVFusion (Liu et al., 2023) is then applied for object detection. As shown in Table 3, the detection mAP of our generated frames remains much closer to that of raw frames, whereas MagicDrive-V2 (Gao et al., 2025) exhibits larger drops. These results demonstrate that our method more effectively preserves key scene elements across weather transformations.

| | ACDC weather training samples | ACDC case 1 | ACDC case 2 | nuScenes case 1 | nuScenes case 2 |
|---|---|---|---|---|---|
| input | | | | | |
| fog | | | | | |
| rain | | | | | |
| snow | | | | | |
| night | | | | | |

Figure 8: Adverse weather transformation results. For ACDC cases, we display the first frame. For nuScenes cases, three key frames are shown to illustrate temporal consistency and visual realism.

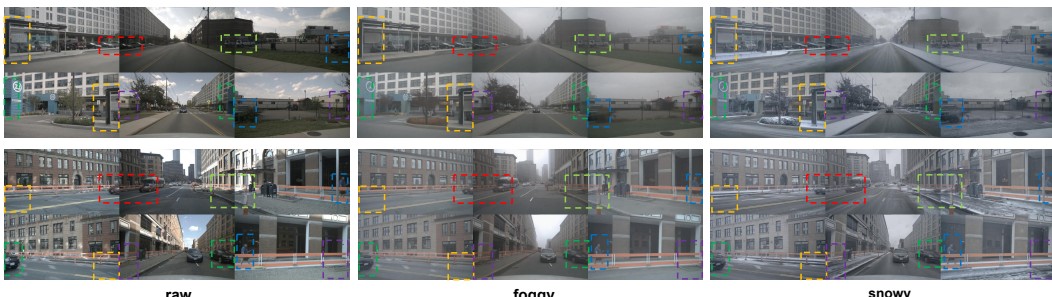

raw        foggy        snowy

Figure 9: Multi-camera results on nuScenes. The dashed boxes with same color indicate same objects in multiple cameras.

## 4.4 QUALITATIVE EVALUATION

**Generation Quality.** To visually compare our method with state-of-the-art approaches, we generate the same nuScenes cases using our model and two leading methods: Vista (Gao et al., 2024) and Panacea (Wen et al., 2024). Results are shown in Figure 6. Our method produces more realistic traffic elements, including vehicles, pedestrians, traffic cones, and lane markings. In comparison, the outputs from Vista and Panacea exhibit noticeable artifacts or less accurate structural details.

**Controllability of Weather.** To show the controllability of our method in generating adverse weathers. Our model is used to transform normal videos to four adverse conditions: foggy, rainy, snowy, and nighttime. Results are shown in Figure 8. For ACDC cases, we show the first generated frame. For nuScenes videos, we show three representative frames to highlight temporal consistency. From it we can see, our method is capable of producing visually realistic and temporally coherent weather effects while preserving the scene structure—such as vehicles, pedestrians, traffic signs and lights.

Notably, even though the nuScenes dataset does not contain any foggy or snowy scenes, our model successfully synthesizes realistic foggy and snowy effects for nuScenes videos, thanks to its generalization ability learned from the ACDC synthesized videos.

## 4.5 MULTI-CAMERA AND LONG VIDEO GENERATION

To assess both cross-view consistency and long-horizon stability, we train a 6-camera model on the nuScenes dataset and apply it to sequences exceeding 250 frames. Representative results are shown in Figure 9, where six synchronized raw camera views and their foggy and snowy translations are displayed. Objects appearing across different cameras remain visually consistent, underscoring the model's ability to maintain strong cross-camera coherence.

Figure 10 presents long-sequence generation, with raw and translated frames at six sampled timestamps. The results reveal no perceptible degradation in visual quality over extended durations.

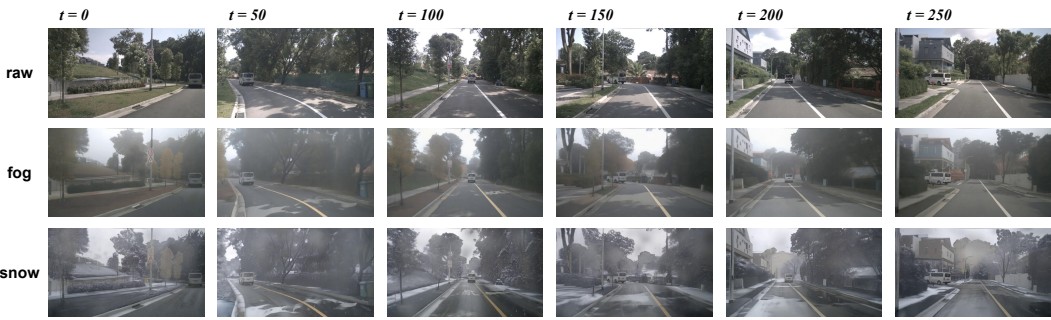

Figure 10: Long video generation results. The results show limited degradation over time, and the white van remains consistent.

Table 3: Comparison with for controllable generation. mAP on origin and generated frames are shown.

| data split | method | mAP↑ | mAP drop↓ |
|---|---|---|---|
| all | Raw | 0.3553 | - |
| | MagicDriveV2 | 0.1817 | 0.1736 |
| | Ours | 0.3376 | 0.0177 |
| rainy | Raw | 0.3435 | - |
| | Ours | 0.2931 | 0.0504 |
| night | Raw | 0.1801 | - |
| | Ours | 0.1645 | 0.0156 |

Table 4: Ablation study on our method.

| Methods | Weather↑ | mAP↑ |
|---|---|---|
| depth control | 1 | 0.5248 |
| sketch control | 1 | 0.5326 |
| lineart control | 0.5611 | 0.6516 |
| separate 3 controls | 0.5328 | 0.6784 |
| concat 3 controls | 0.5444 | 0.6672 |
| masked separate | 0.9586 | 0.6325 |
| masked concat | 0.9567 | 0.6394 |
| *AutoAWG* | 0.9506 | 0.6594 |

Notably, the white van remains coherent and well-preserved across all 250 frames, demonstrating robust temporal consistency.

We further quantify long-horizon stability following Hassan et al. (2025), computing FID and FVD on subsequences of 25, 50, 75, 100, 125, and 150 frames. As shown in Figure 7, our method significantly outperforms Vista (Gao et al., 2024) and GEM (Hassan et al., 2025). Specifically, the score drop between the first and last segments is only 8.4 (FID) and 64.4 (FVD), compared to 19.2 & 208 for GEM and 28.1 & 259 for Vista. These results highlight the superior temporal consistency and overall quality of our approach, even for long video sequences.

For more visual results, please refer to the appendix and our supplementary videos.

### 4.6 ABLATION STUDY

We conduct ablation studies on ACDC samples, each translated into four adverse conditions: fog, rain, snow, and nighttime. The results, summarized in Table 4, highlight the distinct control strengths of different inputs: depth and sketch enhance weather realism but reduce content fidelity, while lineart shows the opposite trend, consistent with the visual analysis in Figure 2.

For multi-control fusion, we compare four strategies: (1) separate encoding, (2) concatenated encoding, (3) mask-based fusion with separate encoding, and (4) mask-based fusion with concatenated encoding. Simple separate or concatenated fusion preserves objects well but yields limited weather realism, resembling lineart control. By contrast, our mask-based fusion strikes a better balance between structure and realism. In particular, the "masked-concat" strategy matches the performance of "masked-separate" while being more efficient, making it the preferred choice. Finally, adding the importance-weighted loss (the *AutoAWG* row) further improves detection accuracy while maintaining visual realism, confirming its effectiveness in enhancing consistency of key objects.

## 5 CONCLUSION

This paper presents *AutoAWG*, a novel framework for generating adverse weather effects in automotive videos.Our method effectively balances the realism of weather transformation with the preservation of critical scene elements. The framework naturally extends to multi-camera configurations which ensuring robust spatiotemporal consistency across synchronized views. Also, it supports open-loop generation of videos with arbitrary lengths, enabling consistent long-duration transformation through progressive conditioning on pre-generated segments. Extensive experiments demonstrate the framework's ability to produce visually realistic, semantically consistent, and temporally coherent transformations under various adverse weather conditions, highlighting its scalability and practical potential for autonomous driving simulation and perception enhancement.

## REPRODUCIBILITY STATEMENT

We are committed to ensuring the reproducibility of our work. The implementation of our method is based on the publicly available `VideoX-Fun` repository[2]. The source code and pretrained models will be shared with reviewers via comments once the discussion forum is opened. Upon acceptance, we will fully release both the code and models to the public.

All datasets used in this work are publicly accessible. Specifically, we use the ACDC dataset[3] and the nuScenes dataset[4]. No proprietary or restricted datasets were employed. The training and evaluation procedures are described in detail within the paper to facilitate replication.

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

# A APPENDIX

## A.1 LARGE LANGUAGE MODELS (LLMS) USAGE DECLARATION

LLMs are used as assist tools to polish our writing.

## A.2 CHOSEN OF LOSS WEIGHT $\alpha$

We perform an ablation study to determine the optimal value of the additinoal loss weight $\alpha$ applied to critical regions. Following the same setup as in the main ablation study, we transform 20 ACDC crop-to-video samples into four adverse conditions, and vary $\alpha$ among {0.5, 1.0, 1.5, 2.0} during training. The results, summarized in Table 5, show that $\alpha = 1.0$ achieves the best balance between weather realism (Weather Score) and content fidelity (Detection mAP), and is therefore used as the default setting.

Table 5: Effect of varying the loss weight $\alpha$ on transformation effects.

| $\alpha$ | Weather Score↑ | Detection mAP↑ |
|---|---|---|
| 0.5 | 0.9508 | 0.6551 |
| 1.0 | 0.9506 | 0.6594 |
| 1.5 | 0.9394 | 0.6628 |
| 2.0 | 0.9139 | 0.6575 |

## A.3 NOTES ON THE EFFECTIVENESS OF MULTI-VIEW STITCHING

In multi-camera video generation, many existing works process each camera independently and introduce cross-attention between adjacent views to enforce consistency (Gao et al., 2023; Wen et al., 2024; Chen et al., 2024). Other methods first generate several non-overlapping views and then condition the generation of the remaining views on those already synthesized (Wang et al., 2024b).

In contrast, our experiments show that such complex mechanisms are unnecessary. A simple strategy—directly stitching all camera views into a single image and feeding it to the video diffusion model—is sufficient to maintain cross-camera coherence.

We attribute this to the inherent design of patch-based DiT architectures. As illustrated in Figure 11, the stitched multi-camera frame is divided into patches and processed jointly by the DiT backbone. The self-attention layers naturally model relationships between patches originating from different cameras, functioning analogously to cross-view attention in multi-view pipelines.

This patchification and self-attention mechanism allows us to avoid explicit multi-view modeling entirely. All experimental results in this paper validate the effectiveness of this simple yet powerful strategy.

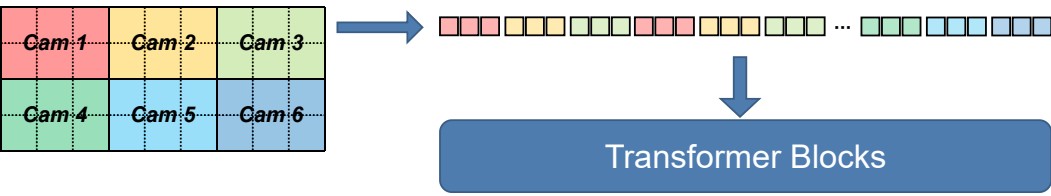

Figure 11: Stitched multi-view frames are split into patches and input to DiT. The self attention of DiT can model their relations.

## A.4 MORE COMPARISON WITH IMAGE-BASED WEATHER TRANSLATION

We further compare our method with existing image-based weather transformation models. Figure 12 and Figure 13 shows the foggy and rainy results generated by GCHQ (Zhao et al., 2024b),

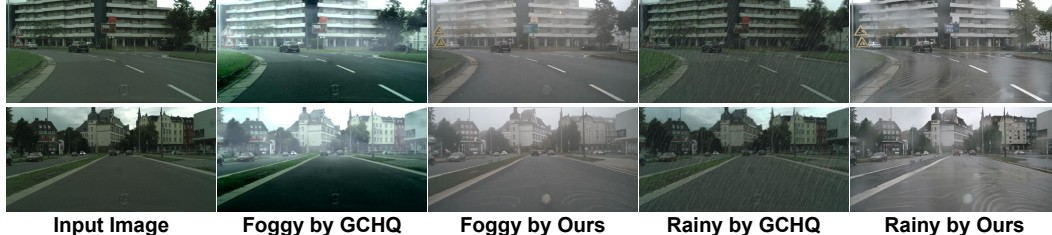

**Input Image** | **Foggy by GCHQ** | **Foggy by Ours** | **Rainy by GCHQ** | **Rainy by Ours**

Figure 12: Visual comparison with GCHQ (Zhao et al., 2024b) for foggy and rainy weather translation. Our method achieves comparable or better realism without dataset-specific training.

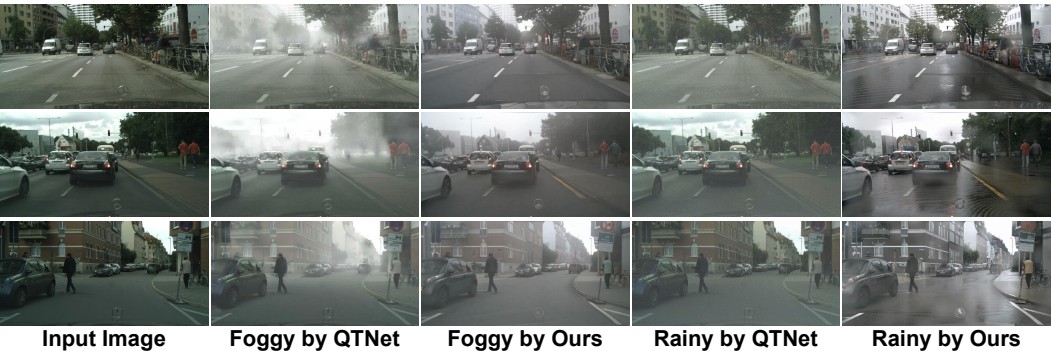

**Input Image** | **Foggy by QTNet** | **Foggy by Ours** | **Rainy by QTNet** | **Rainy by Ours**

Figure 13: Visual comparison with QTNet (Wang et al., 2023) for foggy and rainy weather translation in Cityscapes samples. Our method achieves comparable or better weather effects without dataset-specific training.

QTNet (Wang et al., 2023) and our approach using the same input images, which are chosen from Cityscapes (Cordts et al., 2016). **Despite not being trained on this dataset, our model still produces realistic and high-quality adverse weather effects**, demonstrating comparable or even superior visual fidelity to GCHQ and QTNet. This highlights the strong generalization ability of our model.

## A.5 GENERALIZATION TO OTHER DATASETS

Despite being trained on only a few thousand ACDC images and several hundred nuScenes video clips, our AutoAWG demonstrates strong generalization to unseen datasets. As shown in Figures 12 and 13, the model effectively transfers Cityscapes images to multiple adverse weather conditions. We further validate AutoAWG on the BDD100K dataset (Yu et al., 2020), and Figure 14 illustrates its ability to transform BDD100K video frames as well. These results collectively indicate that AutoAWG achieves robust and reliable cross-dataset generalization, even under limited training data.

## A.6 MORE RESULTS ON MULTI-VIEW AND UNLIMITED-LENGTH GENERATION

Figures 15, 16 and 17 present additional results of our method on multi-camera, long-duration weather transformation. Each input sequence is converted into four adverse conditions: fog, rain, snow, and nighttime. For each weather type, we display representative frames sampled at the 0th, 100th, 200th, and 300th frames to illustrate the temporal progression.

Across all views and frames, the generated sequences maintain high visual quality and consistent object appearance, and no visual degradation is observed over time—even the vehicle behind the ego-car remains visually stable. Moreover, the synthesized weather effects remain photo-realistic throughout the sequence. These results further highlight our model's ability to maintain both high visual quality and strong spatiotemporal consistency across synchronized multi-camera views.

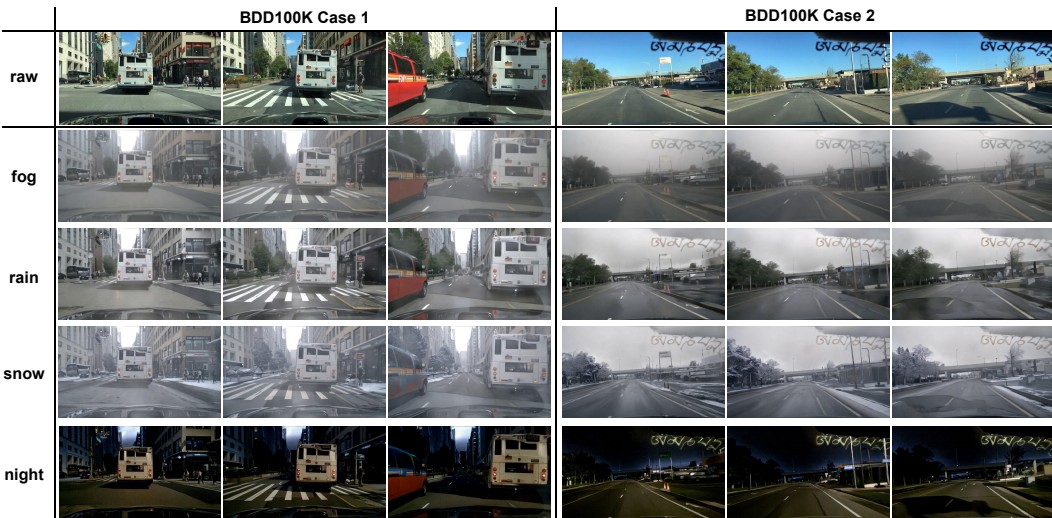

Figure 14: Transferring BDD100K samples to adverse weather using AutoAWG.

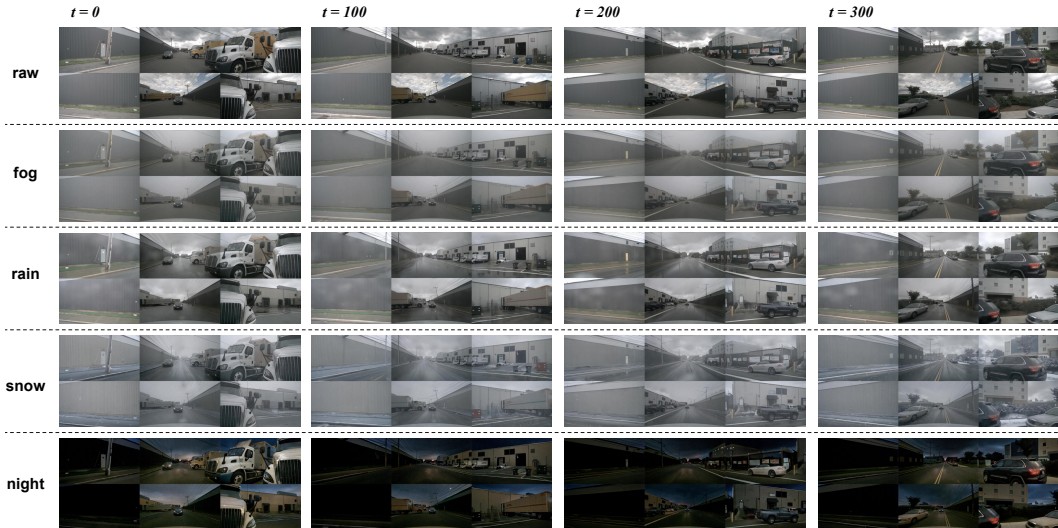

Figure 15: 6-view long-duration weather transformation results on nuScenes.

Notably, our method achieves strong multi-view consistency despite not employing any explicit cross-view attention mechanisms, which are commonly used in prior works (Gao et al., 2023; Wen et al., 2024). We attribute this capability to our model's design: the concatenated multi-view frames are fed as a joint input, allowing the self-attention layers to implicitly learn inter-view correlations during training.

This high-quality generation is further enabled by our use of multiple pixel-level control maps with complementary strengths. Specifically, the lineart map provides strong structural guidance and is capable of generating high-quality results independently. Meanwhile, the depth map delivers finer-grained geometric cues that enhance the realism of weather-specific visual effects. Our adaptive fusion mechanism integrates these heterogeneous controls in a content-aware manner, achieving a robust trade-off between semantic fidelity and stylistic realism.

### A.7 LIMITATIONS AND FUTURE WORKS

Figure 18 illustrates several limitations of our approach, which will be our future works. The limitations can be categorized into four main aspects: (1) lack of controllability over adverse weather

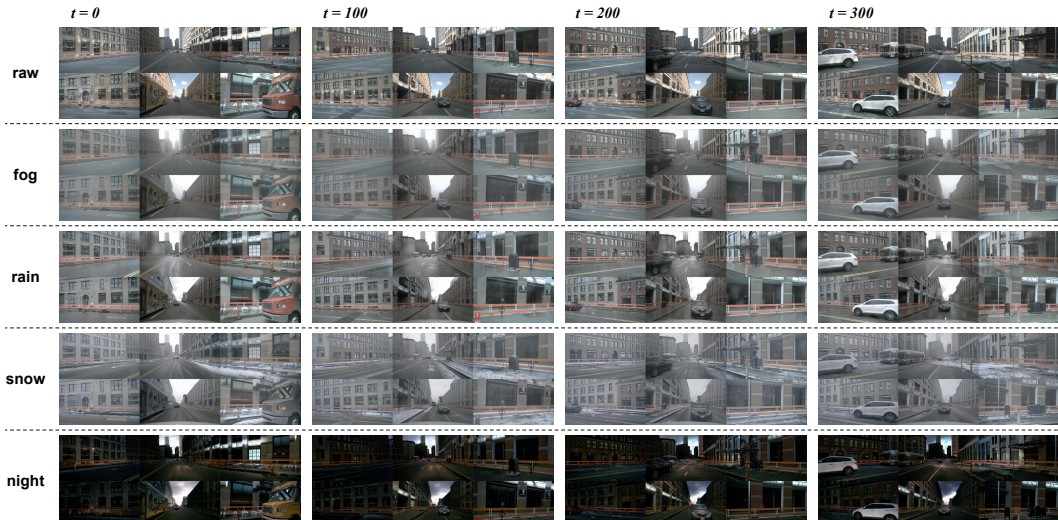

Figure 16: Additional 6-view long-duration weather transformation results on nuScenes.

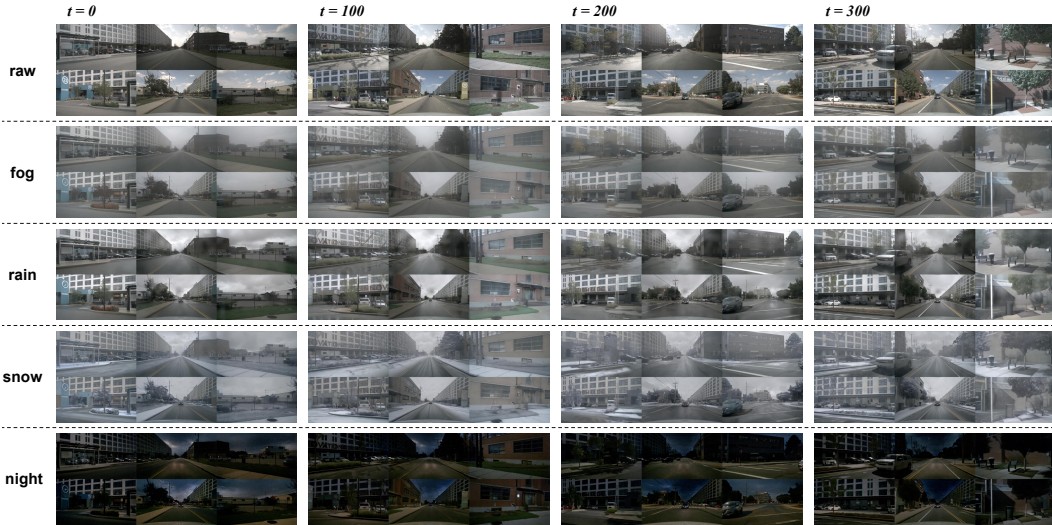

Figure 17: More examples of consistent multi-camera weather transformation results on nuScenes.

intensity, (2) limited capability in composing multiple weather conditions, (3) inability to generate unseen weather types, and (4) ineffectiveness in removing adverse weather.

First, since our model is trained with fixed and simple text prompts without intensity-specific annotations, it struggles to produce weather effects with varying degrees of severity. As shown in Figure 18a, the outputs for both light and heavy fog/snow appear visually similar, lacking noticeable variation in intensity. This limitation, however, could be mitigated by incorporating training data annotated with varying weather intensities.

Second, as illustrated in Figure 18b, our method shows inconsistent behavior when attempting to generate combined weather types. For instance, fog+snow yields a reasonable blend of both styles, while snow+night fails to capture nighttime characteristics.

In fact, the root cause of the above two types of limitations lies in the absence of such scenarios in our training data, and the current training setup does not include prompts involving weather intensities and combinations. For our framework, supporting additional weather intensities or compound weather conditions is fundamentally a data problem rather than a limitation of the model itself. In future work, we plan to enhance these capability by training on more data.

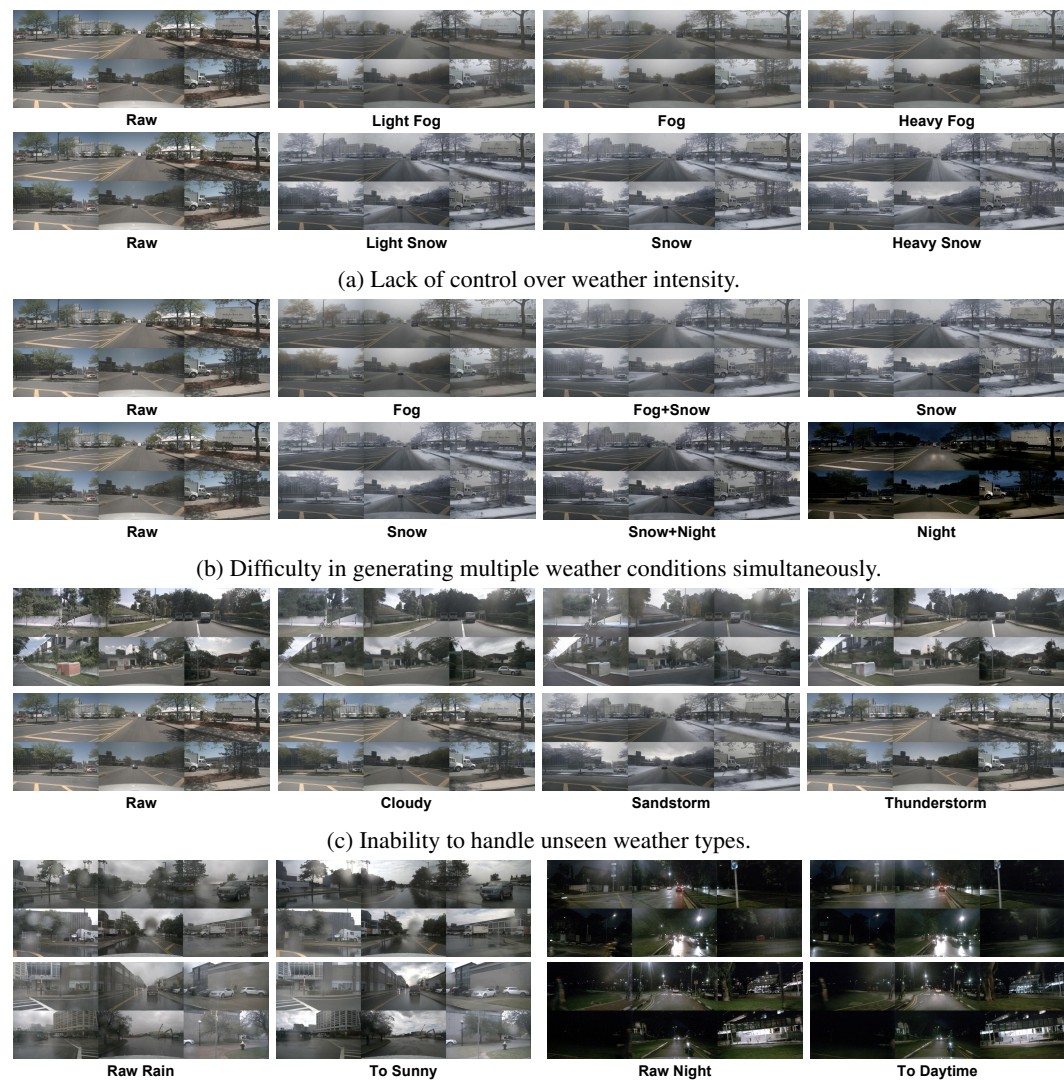

(a) Lack of control over weather intensity.

(b) Difficulty in generating multiple weather conditions simultaneously.

(c) Inability to handle unseen weather types.

(d) Ineffectiveness in removing adverse weather.

Figure 18: Limitations of our method across four aspects: intensity control, unseen weather generalization, multi-weather composition, and weather removal.

Third, Figure 18c shows that our model struggles with generating weather types not present in the training data. While the "cloudy" transformation adds more clouds compared to the original input, the results for "sandstorm" and "thunderstorm" are inaccurate—mistaking sandstorms for snow and failing to depict thunderstorms altogether.

Finally, we explore the reverse task of transforming adverse weather into clear conditions. As shown in Figure 18d, while our model can partially reduce rain effects and brighten the rainy scenes, residual reflections on the road remain. For nighttime inputs, no clear transition to daytime is achieved, although the outputs exhibit reduced noise compared to the original nighttime frames. We think this is due to the inaccurate control maps and segmentation masks in such adverse weather conditions.

In addition, similar to many existing approaches (Gao et al., 2023; Wen et al., 2024), our method does not explicitly model real physical dynamics, which may lead to phenomena that violate physical laws—for example, water on the road not being splashed, or accumulated snow not being compressed by vehicles. However, such artifacts do not affect the stability of downstream perception algorithms. As with many world-modeling approaches, incorporating physically grounded weather dynamics is a promising direction for future work.