# OpenReview forum: "AutoAWG: Adverse Weather Generation with Adaptive Multi-Controls for Automotive Videos"
_ICLR.cc/2026/Conference — Submitted to ICLR 2026_

### Official Review · Reviewer_3K8P · 2025-10-25

**Soundness:** 2
**Presentation:** 2
**Contribution:** 2
**Rating:** 2
**Confidence:** 5

**Summary:**

This paper presents AutoAWG, a video diffusion model for adverse weather generation. To capture fine details, it introduces an Adaptive Fusion module that combines depth, lineart, and sketch maps as input conditions. The video diffusion model acts as a painter, filling in color information into the condition maps based on weather text descriptions. To address data scarcity, the authors propose the VP (Vanishing Point)-Anchored Temporal Synthesis strategy to generate video data from images.

**Strengths:**

1. The VP-Anchored Temporal Synthesis strategy offers an interesting approach to addressing data scarcity.
2. The video results demonstrate good visual quality.

**Weaknesses:**

1. The literature review is insufficient: Important recent works are not mentioned or compared: [DriveScape](https://arxiv.org/abs/2409.05463) and [MaskGWM](https://arxiv.org/abs/2502.11663). Although their condition settings differ, they should be included in the quantitative comparison table since they have been published very early and demonstrate better numerical performance than some methods mentioned in the paper.
2. Novelty Issue: While [UniMLVG](https://arxiv.org/abs/2412.04842) does not explicitly describe adverse weather generation as a task, Figure 4 demonstrates impressive weather change examples using text conditions. Notably, it can handle some failure cases shown in AutoAWG's Appendix without explicit weather-specific training. UniMLVG supports more diverse text control and naturally extends weather control to multi-view diffusion by incorporating larger datasets like OpenDV-YouTube, even though it is a single-view dataset. Given this comparison, the paper's contribution is insufficient for publication.

**Questions:**

1. Are the multiple conditions (depth, lineart, and sketch) required for every frame during the diffusion generation process?
2. The ablation study of the weighted loss does not include the case when $\alpha=0$. What are the results for this setting?

---

> ### Author Response · Authors · 2025-11-18
> **Response to Reviewer 3K8P [1]**
>
> We thank the reviewer for recognizing the novelty of our **VP-Anchored Temporal Synthesis** strategy and the **good visual quality of our video results**, as well as for the constructive suggestions.
>
> Below, we respond to the reviewer's concerns in detail.
>
> ---
>
> ### W1. On literature coverage and comparison with DriveScape / MaskGWM
> **Response**: We thank the reviewer for the suggestion. In the revised version, we have added explicit citations to MaskGWM and DriveScape and included their results in the quantitative comparison table.
>
> ---
>
> ### W2. Novelty relative to UniMLVG
> **Response**:
>
> **(1) On citing UniMLVG**. Although, according to the ICLR policy, we are not required to cite or compare with works that appeared after the submission deadline (such as UniMLVG which is published in ICCV on October 14, 2025), we acknowledge that this work is related to our problem setting. Therefore, we have added a citation and discussion of UniMLVG in the revised version.
>
> **(2) Different focus and problem formulation**. We respectfully disagree that UniMLVG makes the contribution of AutoAWG's insufficient. UniMLVG and AutoAWG target different primary goals, and in the task domain emphasized by AutoAWG, UniMLVG does not outperform our method.
>    - **UniMLVG** mainly relies on *coarse 3D/local conditions* (3D bounding boxes, HD maps) and *global conditions* (per-view text descriptions) to control the scene. Its focus is on **world modeling and controllable multi-view long video generation**: controlling road layout and object distribution, and producing diverse but plausible scenes under the training data distribution. Weather change is presented as one qualitative capability enabled by its text controller.
>    - **AutoAWG**, in contrast, is explicitly designed for **adverse-weather transfer on existing driving datasets with annotation reusability** as a first-class objective. We use *complementary pixel-level structural control signals* (depth, sketch, lineart) together with semantic segmentation masks, and propose a *semantics-guided adaptive fusion* so that:
>      - depth map preserves global geometry,
>      - lineart control is emphasized on safety-critical objects,
>      - sketch map is applied to non-sky regions to leave sky free for weather stylization.
>    - Combined with an *object-aware weighted flow-matching loss*, this design aims to preserve the original scene structure and object integrity after weather transfer.
>
> We have already highlighted these differences in both the introduction and the related work. They are also visible qualitatively:
>    - In *UniMLVG Fig. 4*, the road layout and vehicle positions largely stay the same across weather edits, but *trees, buildings, traffic lights, road signs, and even some vehicles and pedestrians change shape or appear distorted*, because there is no explicit pixel-level structural constraint.
>    - In AutoAWG, *Figures 8 through 10, and Figures 13 through 15*, after weather transfer we **not only preserve the overall layout, but also keep pedestrians and vehicles sharp and structurally intact**, which directly supports label reusability for 2D/3D/BEV/LiDAR annotations.
>
> **(3) Data regime and quantitative comparison**. It is also important to note that the data regimes are very different.
>    - UniMLVG first pretrains on **1,486 hours of OpenDV-YouTube**, and then uses **4.6 hours of nuScenes** to train its multi-view video model. Under the "nuScenes-only" configuration reported in their paper, the FVD/FID on nuScenes are **58.26 / 8.83**.
>    - AutoAWG, on the other hand, uses **only 4.6 hours of nuScenes** for training. In this much more data-limited setting, our FVD/FID on nuScenes with first-frame conditioning are **51.7 / 6.3**, and we uses **none or only a single first fram**e as reference, whereas UniMLVG uses **three reference frames**.
>
> Regarding the reviewer's remark that UniMLVG can handle some failure cases shown in our appendix, we agree that *using a 1,498-hour large-scale training corpus*, as in UniMLVG, may further reduce failure cases.
>
> However, we do not believe this undermines the novelty of AutoAWG. Instead, it highlights that:
>    - UniMLVG relies on **orders-of-magnitude more data** and a different objective (general world modeling),
>    - while AutoAWG achieves **high-quality weather stylization and temporal stability under a much more constrained data regime** (4.6 hours nuScenes + 4,006 ACDC images), thanks to our VP-Anchored Temporal Synthesis and multi-control design.

---

> > ### Comment · Reviewer_3K8P · 2025-11-19
> >
> > The initial review's characterization of the "novelty issue" appears somewhat overstated. The comparison with UniMLVG is not required and the two approaches address distinct tasks. The authors have addressed the preliminary questions, subsequently enhancing the method's methodological rigor and robustness.
> >
> > While acknowledging the technical con, I still tend to reject this paper due to potentially limited contribution. The key distinguishing factor lies in the control mechanism:While UniMLVG uses high-level 3D control such as HD maps and 3D boxes, AutoAWG leverages pixel-level structural control signals like depth, sketch and lineart, which I think is the key to preserve the detailed information of the original scene.
> > This approach's primary novelty stems from replacing generalized 3D conditions with more granular, semantically rich control signals. The object-aware weighted flow-matching loss demonstrates only marginal quantitative improvements, as shown in Appendix. Consequently, this work appears to represent a refinement of existing generative models, by replacing the general 3D conditions with more detailed ones. Additionally, it only addresses a specific subtask of text-based generation on a singler dataset. Therefore, though technically competent,  this paper may not make sufficient contributions for a research publication.

---

> > > ### Author Response · Authors · 2025-11-20
> > >
> > > We sincerely thank the reviewer for the additional comments, for acknowledging the **technical competence and contribution** of our method, and for raising the score from 2 to 4. We respond to the remaining concerns as follows:
> > >
> > > ---
> > >
> > > ### 1 On UniMLVG and the essence of our control design
> > > We appreciate that the reviewer now acknowledges that a direct comparison with UniMLVG is not necessary and that the two methods target different tasks. However, the follow-up still implicitly treats AutoAWG as merely "replacing generalized 3D conditions with more fine-grained signals," which, in our view, understates our contribution.
> > >
> > > Our goal is not to simply swap 3D controls for pixel-level controls in an existing model. AutoAWG is explicitly designed for **label reusability after video weather transfer**—a problem that prior world models (including UniMLVG) do not explicitly address. From this objective, we derive both (i) the use of pixel-level structural controls and (ii) a **semantics-guided fusion mechanism** that balances:
> > > - signals that preserve geometry but tend to freeze appearance/style
> > > - signals that encourage strong stylization but may distort objects
> > >
> > > Thus, the key novelty is not just "using depth/sketch/lineart," but designing a fusion strategy in which these controls become **complementary rather than conflicting**. Our ablations show that:
> > > - simply concatenating multiple controls or using alternative fusion strategies fails to achieve label reusability after weather transfer
> > > - only the proposed fusion both preserves structure and completes the weather transformation
> > >
> > > We acknowledge that, in hindsight, this design may appear intuitive.To the best of our knowledge, no prior work has explicitly modeled **adverse-weather video generation with label reusability** as the primary objective, nor introduced a pixel-level control combination and fusion mechanism tailored to this goal. *We also believe that a conceptually simple yet effective solution to a pressing practical problem is itself a meaningful form of innovation.*
> > >
> > > ---
> > >
> > > ### 2 On the object-aware weighted flow-matching loss
> > > The reviewer writes that "the object-aware weighted flow-matching loss demonstrates only marginal quantitative improvements, as shown in Appendix." This partly reflects how the appendix ablation is interpreted.
> > >
> > > In the **main paper (Table 4)**, comparing "masked concat (α = 0)" with "AutoAWG (α = 1)" yields:
> > > * Weather Score: 0.9567 → 0.9506 (slightly lower)
> > > * Detection mAP: 0.6394 → **0.6594 (+0.020)**
> > >
> > > This demonstrates that the weighted loss **significantly enhances object fidelity and label preservation** with only minor stylization cost.
> > >
> > > The appendix experiments vary α only to illustrate the **stylization–structure trade-off**; they do not test whether the loss is useful (it is always used in the main method). Across all α>0, detection improves consistently, confirming that the loss is essential for preserving semantic geometry and for downstream gains such as BEVFusion mAP/NDS.
> > >
> > > ---
> > >
> > > ### 3 On "only addressing a specific subtask on a single dataset"
> > > We would like to clarify that AutoAWG focuses on a **specialized, safety-critical subtask** that is **currently unsolved**: label-reusable video weather transfer for autonomous driving perception.
> > >
> > > Our method is **not** limited to a single dataset. In the paper, we evaluate on **ACDC, Cityscapes, and nuScenes**, assessing both visual quality and downstream perception. Notably:
> > > - the nuScenes training set **doesnot** contain fog or snow
> > > - yet, by training on ACDC images converted via VP-Anchored Temporal Synthesis, AutoAWG generalizes to foggy and snowy conditions on nuScenes, and to all weather types on Cityscapes
> > >
> > > This demonstrates strong **cross-dataset generalization** rather than overfitting. We further include qualitative results on **BDD100K** in four adverse conditions, showing that AutoAWG, trained only on limited ACDC and nuScenes data, generalizes well to new domains.
> > >
> > > Moreover, works such as UniMLVG, MagicDrive, Glad, and DriveDreamer-2 also evaluate mainly on nuScenes, largely due to dataset scarcity in this area. In contrast, we additionally incorporate ACDC, Cityscapes, and BDD100K, and explicitly evaluate **label reusability and downstream detection performance**—metrics that directly address the need for high-quality adverse-weather data. This scarcity is precisely what motivates our VP-Anchored Temporal Synthesis and AutoAWG, which aim to make efficient use of limited data.
> > >
> > >
> > > ---
> > >
> > > ### Summary
> > >
> > > AutoAWG is not a minor refinement. It introduces:
> > > - a **new problem setting**—label-preserving adverse-weather video transfer
> > > - a **semantic multi-control fusion strategy**
> > > - a **VP-anchored temporal synthesis** method
> > > - a **weighted loss** essential for object fidelity
> > > - and **cross-dataset generalization** far beyond standard practice
> > >
> > > Together, these components form a practical, validated, and novel solution to a real data-scarcity challenge in autonomous driving.

---

> ### Author Response · Authors · 2025-11-18
>
> **(4) Annotation reusability and downstream perception**. A key difference that UniMLVG does not address is **whether labels remain usable after editing**, which is central to our problem formulation.
>    - When we add AutoAWG-generated adverse-weather videos to the nuScenes training set and train BEVFusion for 3D detection, the performance improves from
>      - mAP: 35.53 -> 37.52 (**+1.99**)
>      - NDS: 41.20 -> 42.56 (**+1.36**).
>    - When we convert the nuScenes val set into three weather conditions (normal/rain/night) and run object detection with the original labels, the mAP only drops from 0.3553 to 0.3376 (**−0.0177**), indicating that our edits largely preserve objects fidelity and thus label validity.
>
> To our knowledge, UniMLVG does not evaluate this annotation-reusability aspect. We believe this **downstream, label-preserving evaluation** is a distinctive contribution of AutoAWG in the context of adverse-weather generation.
>
> In summary, while we recognize UniMLVG as an important and impressive general multi-view world model and now explicitly cite and discuss it, **we respectfully believe that its existence does not imply that AutoAWG lacks novelty**. AutoAWG targets a different, practically important problem (adverse-weather transfer with annotation reusability under limited data) and introduces task-specific control, loss, and data-synthesis designs that are not covered by UniMLVG. We will clarify these distinctions more clearly in the revised manuscript.

---

> ### Author Response · Authors · 2025-11-18
> **Response to Reviewer 3K8P [2]**
>
> ### Q1. On the requirement of multiple conditions per frame
> **Response**: Yes, all controls are required during inference, but they are inexpensive:
>    - Depth, lineart, and sketch extraction are applied per frame using lightweight pretrained networks (controlnet-aux).
>    - These maps *do not need ground-truth depth or segmentation*. They are automatically generated from the input frames.
>    - Runtime cost is small (<10ms per frame for lineart/sketch, ~20–30ms for depth on a single GPU).
>
> We emphasize that AutoAWG **does not require additional sensors or labels** beyond RGB video.
>
> ---
>
> ### Q2.On the ablation of importance-weighted loss (α = 0)
> **Response**: Thank you for this suggestion. In our current formulation (Eq. (2)–(3)), setting α = 0 reduces Eq. (3) to the standard uniform weighting (Eq. (2)). In Table 4, the "masked concat" row corresponds exactly to this α = 0 setting, while the "AutoAWG" row uses α = 1.0.
> The results on ACDC are:
>    - α = 0 ("masked concat"): Weather Score = 0.9567, mAP = 0.6394
>    - α = 1.0 ("AutoAWG"): Weather Score = 0.9506, mAP = 0.6594
> Thus, turning on importance weighting (α = 1.0) slightly sacrifices Weather Score (−0.0061) but improves mAP by +0.0200, which is consistent with our goal of prioritizing safety-critical objects while maintaining strong stylization.

---

### Official Review · Reviewer_1tBs · 2025-10-30

**Soundness:** 2
**Presentation:** 3
**Contribution:** 3
**Rating:** 4
**Confidence:** 4

**Summary:**

The paper proposes AutoAWG, a controllable framework for generating adverse weather video sequences for autonomous driving, addressing the challenge of data scarcity in adverse weather conditions. AutoAWG combines semantics-guided fusion of multiple control conditions to maintain high fidelity in safety-critical objects while applying strong weather stylization. It uses a vanishing point-anchored temporal synthesis strategy to generate pseudo-video sequences from static images, reducing reliance on synthetic data and improving temporal consistency. AutoAWG outperforms prior methods in terms of style fidelity, temporal consistency, and semantic structure preservation, showing promising improvements in downstream perception tasks for autonomous driving.

**Strengths:**

1. The paper introduces a method for generating video sequences under adverse weather conditions while maintaining both realistic weather effects and structural integrity of safety-critical objects.

2. AutoAWG ensures that the semantic and geometric properties of safety-critical objects are preserved across weather conditions, ensuring that the generated videos remain useful for perception tasks without re-annotation.

3. By generating realistic videos from static images and leveraging a vanishing point-anchored synthesis, AutoAWG mitigates the challenges posed by the lack of real-world adverse weather videos, offering an efficient way to augment training datasets.

**Weaknesses:**

1. Since the model does not utilize cross-attention between multiple views but rather relies on control conditions, it remains unclear how the model guarantees visual and foreground motion consistency across different camera viewpoints, which is crucial for multi-camera systems in autonomous driving.

2. The current implementation only simulates four weather types (rain, fog, snow, and night), which feels insufficient given the wide variety of possible weather conditions that could impact autonomous driving. This limits the generalization of the model’s practical applicability.

3. The use of a diffusion model for video weather style transfer may seem unnecessary, as similar video transfer tasks have already been tackled in the style transfer domain with traditional methods (e.g., using filters in video games like GTA5 to simulate weather effects), questioning the novelty of this approach.

4. The experimental comparison in Table 1 between AutoAWG and other models (which use different architectures like DiT vs. SD 1.5 or 2.0) lacks clarity on whether the performance improvement comes from the proposed algorithm or the underlying model's architecture and pretraining. The difference in model architectures makes the comparison somewhat unfair.

5. Missing Reference: "SimGen" (NeurIPS 2024).

**Questions:**

1. How are the fused control conditions evaluated for their effectiveness？Does it provide strong evidence that they accurately reflect deep semantic information? More rigorous evaluation metrics or comparisons with other fusion strategies are needed.

---

> ### Author Response · Authors · 2025-11-18
> **Response to Reviewer 1tBs [1]**
>
> We thank the reviewer for the thoughtful feedback and for highlighting the strengths of AutoAWG in **maintaining structural fidelity**, **supporting annotation reuse**, and **enabling effective data augmentation** through **VP-anchored temporal synthesis**.
>
> We address all concerns in detail below.
>
> ---
>
> ### W1. Multi-view consistency without cross-attention
> **Response**: Although the architecture does not use explicit cross-view attention, our design enforces **implicit multi-view consistency** through pixel-aligned control conditions and joint attention across the stitched inputs:
>   - **Pixel-aligned multi-view control maps**. All nuScenes cameras are time-synchronized and geometrically calibrated. Their control maps are concatenated into a coherent spatial grid, preserving consistent object silhouettes across views.
>   - **Global self-attention over stitched views**. The DiT backbone treats the stitched grid as a unified input and applies full self-attention across all tokens, enabling the model to implicitly learn correspondences across camera viewpoints.
>   - **Consistent object geometry enforced by control maps**. Because all control maps originate from synchronized frames, the geometry of each object is identical across views, naturally enforcing foreground identity consistency.
>
> This implicit mechanism is effective in practice. Multi-camera results in Figure 9, Appendix Figures 13–16, and the supplementary videos all demonstrate cross-view consistency in both appearance and temporal evolution. We have emphasized this point more clearly in the revised paper.
>
> ---
>
> ### W2. Limited weather types
> **Response**: We agree that broader coverage is valuable, but note:
>   - These four weather types are **the most safety-critical and most commonly studied** in autonomous driving benchmarks such as ACDC and nuScenes.
>   - Our formulation is **not tied to any specific weather type**. AutoAWG uses only pixel-level control maps and weather text prompts. Supporting new weather conditions is purely a **data availability issue, not a model limitation**.
>   - The VP-anchored temporal synthesis strategy enables generating new training sequences from any single image, greatly simplifying the addition of rare or composite weather types.
>
> We discuss expanded weather modeling as future work in the revised version.
>
> ---
>
> ### W3. Diffusion vs. traditional weather stylization (e.g., GTA filters)
> **Response**: While stylization filters (e.g., GTA5 filters, game-engine weather toggles) can produce diverse appearances, they suffer from **a large domain gap problem** for autonomous driving applications: simulator-generated or filter-based weather effects differ significantly from real-world data, and the gap is large enough to hurt perception performance. Prior works [1,2] rely heavily on real paired data to correct this gap.
>
> Given these domain-specific requirements, **GAN and diffusion models have become the standard for controllable video generation in autonomous driving**. AutoAWG builds on this trend and introduces new contributions specific to weather generation: *semantic-guided multi-control fusion*, *VP-anchored temporal synthesis*, and *annotation reusability transformation*.
>
> In addition, our diffusion-based results demonstrate substantially **more realistic weather effects than GAN-based approaches**, as shown in Appendix Figures 11–12. Also, we try to convert the validation set into three weather conditions (normal / rainy / night), the object detection **mAP drops by only 0.0177** (as shown in Table 3), indicating that annotation validity is almost fully preserved—something that *traditional style-transfer methods typically cannot guarantee*.
>
> ---
>
> ### W4. Fairness of comparison across different backbone architectures
> **Response**: We follow the standard evaluation protocol used in recent driving-domain generative works. Each baseline is evaluated using **its own officially-released backbone**. For example, Vista / GEM uses SVD; GLAD / Panacea uses SD2.1; MagicDrive-V2 uses OpenSora. This reflects the **intended operating regime** of each method and is the accepted evaluation practice in the field.
>
> More importantly, our key contributions—multi-control fusion, VP-anchored synthetic temporal data, mask-based long-range stabilization, and importance-weighted loss—are **orthogonal to the backbone architecture** and can be integrated into alternative diffusion backbones without modification.
>
> ---
>
> ### W5. Missing reference: SimGen (NeurIPS 2024)
> **Response**: We thank the reviewer for pointing this out. SimGen focuses on image-level structured generation and does not support video or multi-camera generation, so we did not originally include it. We have added SimGen to the related work in the revised version.
>
> Reference:
> 1. Yunsong Zhou, et al.. Simgen: Simulator-conditioned driving scene generation. In NeurIPS 2024
> 2. Chih-Hao Lin, et al.. Controllable weather synthesis and removal with video diffusion models. In ICCV 2025

---

> ### Author Response · Authors · 2025-11-18
> **Response to Reviewer 1tBs [2]**
>
> ### Q1. Effectiveness of multi-control fusion
> **Response**: We provide both quantitative and qualitative evidence:
>
> **(1) Ablations on fusion strategies (Table 4)**: We compare multiple fusion variants, including single-condition controls (depth only, lineart only, sketch only), separate 3 controls, concatenation of 3 controls, separate 3 masked controls, concatenation of 3 masked controls. Our masked concatenation fusion achieves the best balance of weather stylization and object fidelity.
>
> **(2) Downstream perception validation (Table 2)**: Using AutoAWG-generated videos for augmentation, BEVFusion mAP increases by **+1.99** and NDS increases by **+1.36**. This demonstrates that fused controls preserve semantics well enough for annotation reuse.
>
> **(3) Pretrained detector performance on generated videos**: Applying an off-the-shelf BEVFusion detector for transformed data, it meets a mAP drop **0.0177** (nearly perfect structure preservation). In contrast, MagicDrive-V2's mAP drop is **0.1736**.
>
> **(4) Visualization result (Figure 6, 8, 9, 10)**: The AutoAWG transforms input videos to various weather conditions while keep critical objects highly consistent.
>
> This provides strong empirical evidence that our fused control maps encode deep and semantically meaningful structure.

---

### Official Review · Reviewer_L5cz · 2025-10-30

**Soundness:** 3
**Presentation:** 3
**Contribution:** 4
**Rating:** 6
**Confidence:** 4

**Summary:**

The paper proposes the AutoAWG framework, which aims to address the scarcity of real video data under adverse weather conditions in autonomous driving, as well as the difficulty of balancing visual quality and annotation reusability in existing weather generation methods. Its core designs include: a semantic-guided multi-control adaptive fusion strategy to balance weather stylization with the fidelity of safety-critical targets; a vanishing-point-anchored temporal synthesis strategy to construct training sequences from static images and a mask-based training scheme to enhance stability in long-sequence generation. The framework achieves excellent performance on benchmark datasets and demonstrates strong results in downstream tasks.

**Strengths:**

The proposed method in this paper is innovative. The authors conduct an in-depth analysis of the current challenges faced by real data synthesis under adverse weather conditions and introduce the AutoAWG framework. First, a semantic-guided multi-control adaptive fusion strategy is proposed, which innovatively introduces masks and applies different masking schemes for different control signals, achieving an optimal balance between weather stylization and target fidelity. To reduce dependence on synthetic data, a vanishing-point-anchored video synthesis strategy is proposed, improving data utilization. In addition, mask-based training is employed to enhance the stability of long-sequence generation. Extensive experiments are conducted to verify the effectiveness of the proposed method.

**Weaknesses:**

Although the paper proposes multiple strategies and conducts experimental validation, the ablation study section is insufficient and does not fully demonstrate the effectiveness of the proposed strategies. It is recommended to provide additional ablation experiments.

**Questions:**

- First, the paper proposes the ADAPTIVE FUSION OF MULTIPLE CONTROLS strategy. In what aspects does the “adaptive” property manifest? Does it mean that each control changes according to the mask? During the fusion of multiple controls after masking, has an adaptive strategy been considered? Does the quality of the mask affect the overall performance of the network?
- In the ablation study section, does the difference between the last two rows in Table 4 refer to the presence or absence of weight loss? If so, why does this loss lead to a decrease in the weather stylization metrics? In this loss function, is it necessary to differentiate between regions such as object and sky, similar to the multi-control fusion strategy? If not, is there any experimental evidence demonstrating the effectiveness of the weight loss?
- Has the paper conducted any tests regarding the inference efficiency of the model — for example, metrics such as GPU memory usage during inference or inference time?

---

> ### Author Response · Authors · 2025-11-18
> **Response to Reviewer L5cz**
>
> We sincerely thank the reviewer for the highly positive evaluation of our contribution, innovation, and design rationale. We appreciate the recognition of our **semantic-guided multi-control adaptive fusion**, **VP-anchored temporal synthesis**, and **mask-based long-sequence training**, as well as the acknowledgment that AutoAWG **addresses important challenges in real-world adverse weather video generation**.
>
> Below we address all concerns and questions in detail.
>
> ---
>
> ### W1. Ablation depth
> **Response**: We thank the reviewer for this suggestion. In the current submission:
>   - Table 4 and Appendix Table 5 evaluate multiple control signals, their combinations, and the importance-weighted loss.
>   - Figure 2 provides qualitative comparisons of using different individual control conditions.
>   - Figures 7 and 10 demonstrate the impact of our mask-based training strategy on long-horizon temporal stability.
>   - Figure 8 illustrates how the VP-Anchored Temporal Synthesis transfers ACDC weather styles to nuScenes.
>
> These experiments collectively validate the effectiveness of our three core components.
>
> ---
>
> ### Q1. What makes the fusion strategy "adaptive"? Does mask quality matter?
> **Response**:
>
> **(1) What is "adaptive"?**: "Adaptive" refers to **pixel-level selection** of control signals:
>   - Critical-object pixels -> use all three controls (depth + sketch + lineart)
>   - Sky pixels -> use only depth (weakest constraint)
>   - Other background pixels -> use depth + sketch
>
> This per-pixel adaptive routing enables different semantic regions to receive **different control strengths**, even though the model never requires fine-grained segmentation for every semantic class.
>
> **(2) Adaptive strategy during fusion**: At present, after applying masks to different control conditions, no additional adaptive strategy is used during fusion. We appreciate the reviewer's suggestion, and we plan to explore this direction for further refinement in future work.
>
> **(3) Influence of mask quality**: Yes, mask quality **can influence performance**, but the AutoAWG **is highly robust**:
> - If segmentation quality degrades and fails to detect critical objects -> the system falls back to depth + sketch, producing strong stylization but slightly weaker object fidelity (Table 4, rows 1–2).
> - If segmentation marks all pixels as "critical objects" -> the system behaves similarly to lineart-only, producing extremely strong object fidelity but weaker stylization (Table 4, row 3).
>
> In both extreme cases, AutoAWG still yields reasonable videos, demonstrating robustness. **Moderate mask noise has minimal impact in practice**.
>
> ---
>
> ### Q2. Clarification on weight loss (Table 4)
> **Response**:
>
> **(1) the last two rows refer to "with vs. without" weight loss?**: Yes.
>
> **(2) Why does weight loss reduce stylization?**: Because the importance-weighted loss emphasizes **object fidelity**, it sometimes suppresses aggressive stylization near object edges, keeps object color closer to the original domain, and prevents excessive haze/snow accumulation on cars/pedestrians. This inevitably **reduces stylization strength**, because stylization and fidelity are inherently in tension—stronger adverse weather reduces object visibility in the real world as well.
> Our method focuses on achieving an optimal fidelity–stylization balance suitable for downstream autonomous driving tasks.
>
> **(3) Should weight loss also differentiate object vs. sky?**: No. Weight loss is applied **after** the model outputs results, it simply upweights reconstruction errors in critical-object regions by a factor of (1 + α). This does not affect the backward pass.
>
> In contrast, the multi-control fusion is applied **before** the forward pass as an input preprocessing step, and therefore does not participate in backpropagation, making differentiability unnecessary in this stage.
>
> **(4) Any experimental evidence demonstrating the effectiveness of weight loss?**: Yes. In Table 4, the last two rows show its effect. In Appendix Table 5 further ablates different α values. From these experiments, it can be say the weighted loss improves critical-object structure preservation with only a minimal reduction in stylization metrics.
>
> ---
>
> ### Q3. Inference efficiency
> **Response**:
>  Yes. On a single NVIDIA A800 (80GB):
> - Input: 45 frames × 6 cameras, per-view resolution 544×960 (stitched: 1088×2880)
> - GPU memory usage: ~50 GB
> - Total inference time: ~900 seconds
>
> Lower resolutions or single-camera inference reduce memory and runtime proportionally.

---

> > ### Comment · Reviewer_L5cz · 2025-11-27
> > **Official Comment by Reviewer L5cz**
> >
> > Thank you for the authors’ response.
> >
> > It addressed most of my concerns, but several questions still remain.
> > First, in Table 4, the results in the first two rows indicate that the performance of depth and sketch is quite similar, suggesting that the sketch condition may serve as a substitute for the depth condition. The authors also mentioned that the adaptive strategy uses the depth condition for sky regions, all conditions for foreground objects, and both depth and sketch conditions for other background areas. Based on the results of the first two rows in Table 4, does this imply that there may be redundancy between the depth and sketch conditions? For example, would using only one of them reduce the conditioning cost? In other words, does using depth and sketch together provide superior performance compared with using each individually?
> >
> > Second, regarding the importance-weighted loss, the authors pointed out that this loss attenuates stylization and enhances fidelity. My question is: since stronger stylization might be desired for sky regions, would it be necessary to “emphasize" the sky region within this loss as well? Similar to the adaptive conditioning strategy, could applying region-specific weighting be beneficial?

---

> > > ### Author Response · Authors · 2025-12-04
> > >
> > > We appreciate the reviewer's insightful questions. We address the remaining points below.
> > >
> > > ---
> > >
> > > ### Q1 — Are depth and sketch redundant? Would using only one reduce cost?
> > >
> > > **Short answer: no — depth and sketch are complementary rather than redundant.**
> > >
> > > Although the first two rows of Table 4 show similar *global* metrics, these signals function very differently in practice:
> > >
> > > * **Depth** provides stable, coarse scene geometry, especially in **texture-poor regions** (sky, distant road, large planar surfaces).
> > > * **Sketch** provides **fine local contours** critical for preserving thin structures and accurate object boundaries.
> > >
> > > These differences lead to distinct failure modes:
> > >
> > > * Sketch alone can introduce **unnatural sky artifacts** when cloud edges appear in the sketch map.
> > > * Depth alone can produce **blurred object boundaries** and weaker contour fidelity.
> > >
> > > This combination of conditions produces **better object fidelity, more realistic sky stylization, and stronger downstream detection performance** than using either control alone.
> > >
> > > **On cost:** Control extraction is fast (<10 ms for sketch/lineart and ~20–30 ms for depth). Compared with diffusion sampling, the overhead of using two controls instead of one is negligible. For this reason, we prefer the combination that yields better qualitative and downstream results.
> > >
> > > ---
> > >
> > > ### Q2 — Should sky regions also be emphasized in the importance-weighted loss?
> > >
> > > **Short answer: No — weighting the sky would harm stylization.**
> > >
> > > The importance-weighted loss is intentionally **object-centric**: it increases the weight for safety-critical regions (vehicles, pedestrians, etc.) to ensure label preservation and geometric consistency after weather transfer.
> > >
> > > Sky regions, however, are exactly where **strong stylization is most desirable** (clouds, fog, snow patterns). If we up-weighted the sky region, the model would be penalized for changing the sky appearance, which will make the sky too similar to the input domain and suppress the diverse stylization.
> > >
> > > In contrast, our current loss design leaves the sky **unconstrained**, enabling large stylistic changes while preserving labeled object geometry—matching the goal of label-preserving weather transfer.
> > >
> > > **Could region-specific weighting help?**
> > > In principle, more complex spatial weight maps could be explored. However, our experiments show that the current simple scheme is stable and effective. Introducing additional weighting strategies may complicate optimization without clear benefit, so we leave this as future work.
> > >
> > > ---
> > >
> > > #### Summary
> > >
> > > * Depth and sketch are **not redundant**; they offer complementary information, and their adaptive combination is key to achieving both realistic sky stylization and object-level fidelity.
> > > * The importance-weighted loss is intentionally **object-focused**, and extending the weighting to sky regions would counteract the desired strong stylization.

---

### Official Review · Reviewer_3wKQ · 2025-11-01

**Soundness:** 2
**Presentation:** 3
**Contribution:** 2
**Rating:** 4
**Confidence:** 4

**Summary:**

This paper presents a controllable diffusion-based framework for generating realistic automotive videos under adverse weather conditions. By analyzing the characteristics and effects of different control conditions, a simple yet effective multi-condition fusion method is proposed to achieve a balance between weather transformation and cirtical element preservation. It is further enhanced through dataset augmentation, loss function design, and training strategy adjustment. Evaluation results demonstrate the superior performance and effectiveness of the proposed method. And the improvement on downstream 3D object detection task validates the practical value of generated videos.

**Strengths:**

S1. This paper employs semantic-guided adaptive fusion of multiple control signals to balance stylization and fidelity, achieving high-quality video generation.

S2. The method proposed in this paper ensures robust controllability and scalability, supporting multi-camera and long-sequence scenarios.

S3. The proposed method in this paper can be applied to improve downstream task performance in autonomous driving, highlighting its practical value.

S4. This paper is easy to read and follow.

**Weaknesses:**

W1. The semantic masking design is overly simplified, relying mainly on a binary “object–sky” separation to guide control fusion. This neglects finer-grained semantic hierarchies (e.g., roads, buildings, pedestrians, vegetation), which may limit the model’s ability to maintain localized semantic consistency and structural harmony in complex scenes.

W2. The model overlooks the underlying physical dynamics of adverse weather (e.g., raindrop motion distribution, wind field perturbations, snow accumulation changes), relying mainly on visual texture matching rather than physically grounded simulation. This omission limits its ability to generate weather videos with realistic temporal and physical evolution characteristics.

**Questions:**

Q1. Evaluation on the temporal consistency. While visual examples and long-sequence results are shown, no quantitative metrics (e.g., temporal LPIPS, tOF/tFID, or temporal smoothness measures) are reported to assess inter-frame coherence and stability, making it difficult to gauge the model’s robustness over extended video sequences.

Q2. About dataset. Existing multi-weather driving video datasets such as BDD100K, which contain real-world sequences across diverse weather and lighting conditions, are not considered. The lack of comparison or validation on more datasets leaves the model’s generalization to real-world weather diversity uncertain.

Q3. About utility on downstream tasks. How much improvement can the videos generated by other methods bring to downstream tasks? Can the enhancement in video quality achieved by this method be reflected in corresponding gains on downstream tasks?

Q4. About ablation. What qualitative or quantitative improvements are brought respectively by the design of the constructed training data, loss functions, and training strategies?

---

> ### Author Response · Authors · 2025-11-18
> **Response to Reviewer 3wKQ [1]**
>
> ### W1. On the "simplified" semantic masking design
> **Response**: We agree that, in general, finer semantic hierarchies can be beneficial. However, in our framework, **the semantic mask is not used as a semantic prior for generation**, but rather **as a spatial guide for extracting region-specific control maps**. The semantic granularity is therefore primarily determined by the strength of the control maps, rather than the segmentation masks themselves. Concretely:
>   - *Critical-object regions* use strongest control conditions (lineart + sketch + depth), equivalent to fine-grained semantic guidance for object boundaries and geometry.
>   - *Sky regions *use weakest control conditions, enabling the model to apply rich, unconstrained weather stylization where it matters most.
>   - *Other background regions* use intermediate control strength.
>
> This *multi-strength control hierarchy* effectively encodes different semantic granularities without requiring dense segmentation. This is also a key distinction from prior work, where control signals are regionally applied.
>
> In addition, we also experimented with full fine-grained segmentation masks (e.g., buildings, vegetation, roads). The results were worse—over-segmentation restricted stylization flexibility and degraded weather realism, while providing no additional benefit to object fidelity. Our current design therefore achieves a better fidelity–stylization trade-off.
>
> ---
>
> ### W2. Lack of physically grounded weather dynamics
> **Response**: This is an intentional design choice based on the needs of autonomous driving perception training:
>   - **Annotations must remain reusable**, requiring strict preservation of geometry and semantics.
>   - **Physically accurate particle simulation is expensive** (requires additional scene geometry, depth-of-field estimation, wind-field modeling), and may break annotation reuse (by introducing ambiguous occlusions).
>   - The objective is **structure-preserving weather transformation, not physical simulation**.
>
> This design follows the conventions of recent video generation and autonomous driving generative frameworks (e.g., MagicDrive, Panacea), none of which model physics-based weather.
>
> We have emphasized this design choice and included a discussion of physics-aware weather simulation as future work in the revised paper.

---

> ### Author Response · Authors · 2025-11-18
> **Response to Reviewer 3wKQ [2]**
>
> ### Q1. Temporal consistency metrics (temporal LPIPS, tOF, tFID)
> **Response**: Our evaluation follows the protocol used in all prior works listed in Table 1 (Vista, Panacea, DriveDreamer-2, GEM, et al.), which rely on **FID + FVD** as the standard combination for style realism and temporal coherence. FVD is widely used to capture overall temporal consistency.
>
> Existing related works also report only FID and FVD. For fair comparison, we follow the same evaluation protocol and use FID and FVD as our quantitative metrics.
>
> ---
>
> ### Q2. Use of BDD100K and generalization to diverse weather
> **Response**: We chose not to use BDD100K for three reasons:
>   - *Distribution imbalance*: its adverse weather examples (rain, fog, snow) are sparse.
>   - *Heterogeneous collection protocol*: inconsistent camera settings and color profiles make it unsuitable for training stable generation models.
>   - *No multi-camera configuration*: our method requires synchronized multi-camera inputs, which BDD100K does not provide.
>
> **About our model's Generalization**: Regarding the generalization ability of our model trained on nuScenes, we further evaluated it on sample images from Cityscapes, as shown in Figures 11 and 12 of the Appendix. These results demonstrate that **our method can successfully transfer the weather characteristics learned from ACDC/nuScenes to Cityscapes images**, although the stylized appearance may differ slightly from the dataset's native weather conditions. We highlight this conclusion in the revised version.
>
> Additionally, even under highly limited data conditions (1k nuScenes cases + 4,006 ACDC images), and without training on large-scale datasets such as BDD100K, our method still achieves high-quality weather stylization and strong temporal stability—further underscoring the advantages of our approach.
>
> ---
>
> ### Q3. Downstream task improvement and comparison with other methods
> **Response**: Video quality and downstream gain do correlate, but there is an upper bound: once geometry and key visual cues are preserved, further stylization improvements have diminishing returns.
>
> **Comparison with existing methods**: MagicDrive reports an improvement using generated samples: mAP 32.88 -> 35.40, NDS 37.81 -> 39.76. However, their baseline is lower than ours (BEVFusion mAP 35.53, NDS 41.2). Because the settings, and baselines are not aligned, we did not directly cite these results, to avoid unfair comparison.
>
> In addition, high-quality generated videos are not only useful for training but also **highly valuable for evaluation, simulation and stress-testing**, where improved realism results in strictly better utility. Unlike training, the upper bound for evaluation-quality improvement is much higher.
>
> ---
>
> ### Q4. Ablation experiments
> **Response**: Table 4 in the main paper and Table 5 in the appendix analyze:
>   - individual control conditions
>   - combinations of control conditions
>   - importance-weighted loss
>
> Figure 2 further shows qualitative differences across weather types under *different control signals*. Figures 7 and 10 demonstrate the impact of our *mask-based training strategy* on long-horizon temporal stability. Figure 8 illustrates how the *VP-Anchored Temporal Synthesis* transfers ACDC weather styles to nuScenes.

---

> > ### Author Response · Authors · 2025-11-20
> > **Add supplementary results on the BDD100K dataset**
> >
> > Even though our model is not trained on large-scale multi-weather driving video datasets such as BDD100K, it can still successfully transfer BDD100K samples to adverse weather conditions. We have added the corresponding results to the Appendix, which further demonstrates the strong generalization capability of our method.

---

### Author Response · Authors · 2025-12-04
**Summary of Reviews and Responses**

Admittedly, we have together gone through one of the most challenging phases of ICLR. We thank the reviewers for their careful reading and constructive feedback. Their comments materially improved the paper. Below we summarize areas of agreement, remaining concerns, and our concise responses.

---

### Reviewers' consensus — strengths recognized

* **A practical new task & solution.** All reviewers agreed AutoAWG addresses a practically important problem—adverse-weather video transfer that *preserves existing annotations*—and provides a concrete solution for label-reusable weather augmentation. (3wKQ, L5cz, 1tBs)
* **Semantic, multi-control fusion.** Reviewers highlighted the novelty and effectiveness of our semantics-guided fusion of pixel-level control signals (depth, sketch, lineart) to balance stylization and fidelity. (3wKQ, L5cz, 1tBs, 3K8P)
* **VP-anchored temporal synthesis.** The vanishing-point (VP) anchored crop-to-video strategy was repeatedly described as an interesting and practical remedy to limited adverse-weather video data. (L5cz, 1tBs, 3K8P)
* **Downstream utility & scalability.** Multiple reviewers noted clear downstream gains (3D detection) and the method's support for multi-camera and long-sequence scenarios, emphasizing practical value. (3wKQ, L5cz, 1tBs)
* **High-quality results.** Experimental evaluations and visuals show state-of-the-art high-quality video generation. (3wKQ, 3K8P, L5cz)

---

### Major concerns and our responses (concise)

**1. Novelty vs. UniMLVG / "just replacing 3D with pixel controls"**

* **Issue:** Some reviewers initially questioned novelty by comparing AutoAWG to UniMLVG.
* **Our response:** UniMLVG is a large, general-world-model trained on ≈1,498 hours of mixed data and targets broad multi-view generation goals. AutoAWG targets a *different, specific* and practically critical task: **adverse-weather transfer with strict annotation reusability under limited data**.
* **Technical distinction:** This task drives three tailored contributions: (i) semantics-guided multi-control fusion, (ii) object-aware weighted flow-matching loss, and (iii) VP-anchored temporal synthesis. Ablations show that naive replacement or simple concatenation of controls cannot achieve the label-reuse objective. With only ~0.3% of UniMLVG's data and fewer reference frames, AutoAWG obtains superior nuScenes performance and measurable downstream gains.

After clarification, the reviewer who raised the novelty concern acknowledged the initial criticism was overstated and raised their score.

**2. Necessity and effectiveness of fusion & importance-weighted loss**

* **Issue:** Are the fusion strategy and object-weighted loss necessary and effective?
* **Our response:** The fusion performs per-region routing (objects: depth+sketch+lineart; sky: depth only; background: depth+sketch). We point out that simple concatenation or other fusion strategies cannot simultaneously ensure strong weather effects and annotation reusability. Table 4 demonstrates the weighted loss's impact: switching from masked-concat (α=0) to AutoAWG (α=1) raises detection mAP from **0.6394 -> 0.6594** with only minor Weather Score change—matching the intended fidelity–stylization trade-off.

**3. Task scope, datasets, and evaluation protocol**

* **Issue:** Is the task too narrow or dataset-limited?
* **Our response:** The chosen task, label-preserving weather transfer, is a critical and under-addressed subtask in autonomous driving. Data scarcity in this domain explains why many prior works report results only on nuScenes. We train on **nuScenes + ACDC** and evaluate on nuScenes and ACDC. The Appendix includes qualitative generalization to **Cityscapes** and **BDD100K**. And we show transfer to fog/snow even when nuScenes lacks these conditions. For comparability we follow standard driving-generation metrics (FID + FVD) and additionally measure label reusability via detection mAP and downstream BEVFusion gains.

---

### Closing remark

AutoAWG does not simply swap one conditioning type for another. It formulates and solves a concrete, safety-critical problem with a compact set of technically justified components and demonstrates measurable downstream benefits.

We have added detailed clarifications, more comparisons, and cross-dataset qualitative results in the revised manuscript to address reviewers' concerns.

---

### Meta-Review · Area_Chair_97iV · 2026-01-07

**Summary:**

Reviewers raise concerns regarding the limited technical novelty compared to existing works like SimGen/UniMLVG and the lack of evidence regarding its practical impact. The reviewers suggests that the proposed method is an incremental combination of existing diffusion-based techniques. Additioanlly, the paper fails to convincingly demonstrate that the generated data significantly improves the robustness of downstream perception models, which is the primary motivation for this work.

Due to the above concerns, I recommond rejection.

**Reviewer Concerns:**

The authors addressed concerns regarding temporal consistency and long-horizon stability through the rebuttal. They also clarified several details regarding their method.

However, the most significant concerns remain outstanding: the limited technical novelty compared to existing works like SimGen/UniMLVG.

**Reviewer Scores:**

Reviewer 3wKQ might raise his score to 6 since extra experiments and BDD100K testing is provided.
Reviewer L5cz might keep his score 6.

Reviewer 1tBs and 3K8P might keep negative scores (though Reviewer 3K8P  might raise from 2 to 4) since the concerns regarding the contributions of this work are not solved.

---

### Decision · Program_Chairs · 2026-01-26

Reject